# Variational Learning Induces Adaptive Label Smoothing

## Abstract

We show that variational learning naturally induces an adaptive label smoothing where label noise is specialized for each example. Such label-smoothing is useful to handle examples with labeling errors and distribution shifts, but designing a good adaptivity strategy is not always easy. We propose to skip this step and simply use the natural adaptivity induced during the optimization of a variational objective. We show empirical results where a variational algorithm called IVON outperforms traditional label smoothing and yields adaptivity strategies similar to those of an existing approach. By connecting Bayesian methods to label smoothing, our work provides a new way to handle overconfident predictions.

## 1. Introduction

Adaptive strategies to Label Smoothing (LS) (Szegedy et al., 2016) aim to adapt the label noise according to the type of data example. Such adaptation can be more effective in practice than its traditional counterpart where the label noise is the same for all examples. Adaptation is useful to handle examples that may have labeling errors, distribution shift, or calibration issues. For such cases, the effectiveness of adaptation has been extensively studied, for example, see Ghoshal et al. (2021); Lee et al. (2022) for generalization improvements, Zhang et al. (2021); Ko et al. (2023) for mis-labelled examples, Park et al. (2023) for miscalibration, and Xu et al. (2024) for out-of-distribution detection. Adaptivity is useful for label smoothing to handle all such cases.

One major problem with adaptive label smoothing is that it is not easy to design a good adaptivity strategy. For example, a simple approach is to adapt the label noise by using model's predictions but there are many ways to do this, for examples, Park et al. (2023) set the noise based on the logits, Zhang et al. (2021); Ko et al. (2023) use the predictive probabilities (obtained with softmax), while Lee et al. (2022) use their entropy. All of these are reasonable ideas but the choice of a good strategy for a given problem is not always straightforward. A strategy that reduces miscalibration may not be most effective for handling outliers or mislabeling. Focusing on one issue at a time has given rise to a lot of

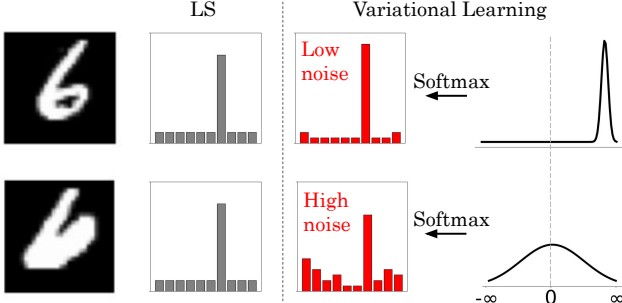

*Figure 1.* Given a regular 6 digit (top) and an atypical one (bottom), Label Smoothing (LS) assigns the same label noise to both (gray bars) while variational learning assigns higher noise to the atypical example (red bars). Adaptivity naturally arise due to the posterior.

ad-hoc and heuristic strategies, and, despite their usefulness, designing an adaptive strategy for a task in hand remains tricky. Our goal here is to simplify the process by presenting and analyzing algorithms that naturally induce adaptivity.

We show that variational learning naturally induces an adaptive label smoothing. The smoothing arises due to the use of the expectation of the loss in the variational objective, taken with respect to the posterior distribution. The expectation gives rise to a label noise (among other types of noises) which is customized for each example through its features. Our key contribution is to derive the exact form of the label noise (Eq. 9) for many problems and study their behavior. We show extensive empirical results analyzing the label noise induced by Improved Variational Online Newton (IVON) (Shen et al., 2024). We show the following

1. Variational learning assigns higher noise to atypical or ambiguous examples (Fig. 1 and Fig. 3).

2. IVON's adaptive label noise behaves similarly to the proposal of Zhang et al. (2021).

3. IVON consistently outperforms Label Smoothing in presence of labeling errors, giving up to 9% accuracy boost for pair-flip noise (Fig. 8) and sometimes even around 50% for data-dependent noise (Fig. 7).

Our work connects label smoothing literature to Bayesian methods, thereby providing a new way to handle overconfident predictions in deep learning.

## 2. Label Smoothing and Adaptivity Strategies

Label Smoothing (LS) is a simple technique where the true label vector $\mathbf{y}_i$ (length $K$) are replaced by a smoothed version. In its simplest form, a convex combination is used where the smoothed labels are defined as

$$\mathbf{y}'_i = (1-\alpha)\mathbf{y}_i + \alpha\mathbf{u}, \tag{1}$$

for some scalar $\alpha \in (0,1)$ with $\mathbf{u}$ as a vector of $1/K$ with $K$ being the number of classes. This simple technique is effective to penalize overconfident predictions because the noise $\alpha(\mathbf{u} - \mathbf{y}_i)$ reduces the importance of the label during training (Pereyra et al., 2017). Multiple works have studied its effectiveness, for example, to improve calibration and representation (Müller et al., 2019), to favor flatter solutions (Damian et al., 2021), and improve robustness to mislabelled data (Lukasik et al., 2020; Liu, 2021) due to its connections to loss correction (Patrini et al., 2017). Despite its simplicity, LS has clear practical advantages.

Adaptive label smoothing aims to inject noise according to the type of data example, for example, during learning, we may want to inject a noise to get the smoothed label

$$\mathbf{y}_{i|t} = \mathbf{y}_i + \boldsymbol{\epsilon}_{i|t}. \tag{2}$$

The noise $\boldsymbol{\epsilon}_{i|t}$ may depend on the current model parameter $\boldsymbol{\theta}_t$ at iteration $t$, and can be varied according the model's opinion regarding the relevance of the examples. Such adaptive label smoothing uses additive noise to reweigh examples during training. Many studies have shown the effectiveness of the adaptive noise, which ranges from improvements in generalization (Ghoshal et al., 2021; Lee et al., 2022), robustness to mislabeled data (Zhang et al., 2021; Ko et al., 2023), improving calibration (Park et al., 2023) and out-of-distribution (OOD) detection (Xu et al., 2024). By adapting label noise, such methods aim to down-weight the problematic examples.

While adaptivity is desirable, it also requires additional effort to design a good strategy to adapt. Each specific issues may require a different type of noise, for instance, what works to reduce miscalibration, may not be most effective for handling OOD detection or mislabling. Focusing on one issue or strategy at a time has given rise to a lot of ad-hoc and heuristic strategies, and, despite their usefulness, clarity of good ways to design adaptivity strategy is lacking.

The simplest approach is to adapt by using the model predictions based on the logits $\mathbf{f}_i(\boldsymbol{\theta}_t)$, but there are many ways to use them. Zhang et al. (2021) use the following update

$$\mathbf{u} \leftarrow \mathbf{u} + \mathcal{S}\left[\mathbf{f}_i(\boldsymbol{\theta}_t)\right], \tag{3}$$

where $\mathcal{S}[\mathbf{f}]$ vector (length $K$) with $j$'th entry defined as

$$\mathcal{S}_j\left[\mathbf{f}\right] = \frac{e^{f_j}}{\sum_{k=1}^{K} e^{f_k}}, \tag{4}$$

although they normalize the $\mathbf{u}$ after every epoch, not at every iteration. A similar rule is used by Ko et al. (2023). Instead of directly using the logits, Lee et al. (2022) use them to adjust $\alpha$. They do so by using the entropy of the model-output distribution, assigning a smaller smoothing to high entropy samples and larger smoothing to low entropy samples. Another approach by Park et al. (2023) decrease the label noise linearly as the logit $\mathbf{f}_i(\boldsymbol{\theta}_t)$ increase. There are multiple ways to use predictions but the choice of a good strategy for a given problem is not always straightforward.

Intuitively, using model's predictions makes sense because predictions can tell us about the relevance of examples. Regions where model is inaccurate may also contain examples that need special attention but also those that are impossible to predict. Some works have explored this from the Bayesian viewpoint, although only using the posterior over the labels. For example, Li et al. (2020) motivate adaptive smoothing using Bayes error rate, implying larger smoothing to example that lie near the decision boundary. Similarly, Ghoshal et al. (2021) use a PAC-Bayes bound to motivate adaptivity. However, there are no approaches investigating the effectiveness of posterior over $\boldsymbol{\theta}$.

In this paper, we show that directly learning the posterior using a variational method natural yields an adaptive label noise. Adaptivity introduced in this fashion directly takes various causes of uncertainty, some of which is then handled through the label noise. The uncertainty in parameter have other desired effect that are often missed when only focusing on the label noise. In our context, this can simplify the design of adaptive label smoothing or may even allow us to entirely skip the step. We will now discuss the adaptive label noise induced by variational learning.

## 3. Variational Learning Induces Adaptive LS

Variational learning aims to optimize for distribution over parameters $\boldsymbol{\theta}$ which is fundamentally different from traditional deep learning where we minimize empirical risk,

$$\bar{\ell}(\boldsymbol{\theta}) = \sum_{i=1}^{N} \ell_i(\boldsymbol{\theta}) + \mathcal{R}_0(\boldsymbol{\theta}), \tag{5}$$

with loss $\ell_i(\boldsymbol{\theta})$ for the $i$'th example in the training dataset. The regularizer $\mathcal{R}_0(\theta)$ is often implicitly defined through various training choices, such as, weight-decay, initialization, and architecture design. In contrast, variational learning aims to find a distribution $q(\boldsymbol{\theta}) \in \mathcal{Q}$ which minimizes

$$\mathcal{L}(q) = \sum_{i=1}^{N} \mathbb{E}_q\left[\ell_i(\boldsymbol{\theta})\right] + \mathbb{D}_{\mathrm{KL}}[q(\boldsymbol{\theta}) \,\|\, p(\boldsymbol{\theta})]. \tag{6}$$

The second term is the Kullback-Leibler (KL) Divergence where the $p(\boldsymbol{\theta}) \propto \exp(-\mathcal{R}_0(\boldsymbol{\theta}))$ can be defined implicitly

similarly to deep learning. Throughout, we will set $q(\boldsymbol{\theta})$ to take Gaussian forms and show that, despite their differences, variational learning can be implicitly seen as minimizing a noisy version of Eq. 5. Existing works have studied the weight-noise (Zhang et al., 2018; Khan et al., 2018) but our goal here is to specifically study its effect on label noise.

### 3.1. A Simple Example: Logistic Regression

We start with logistic regression where we can write a closed-form expression for the adaptive label noise. The result extends to all loss functions using generalized linear model. We will consider all such extensions (including neural networks) afterwards. For now, we consider a loss function for binary labels $y_i \in \{0, 1\}$ with model output $f_i(\boldsymbol{\theta})$,

$$\ell_i(\boldsymbol{\theta}) = -y_i f_i(\boldsymbol{\theta}) + \log\left(1 + e^{f_i(\boldsymbol{\theta})}\right). \quad (7)$$

In logistic regression, we have $f_i(\boldsymbol{\theta}) = \boldsymbol{\phi}_i^\top \boldsymbol{\theta}$ where $\boldsymbol{\phi}_i \in \mathbb{R}^P$ is the feature vector. For simplicity, let us assume $\mathcal{R}_0(\boldsymbol{\theta}) = \frac{1}{2}\|\boldsymbol{\theta}\|^2$ to be a quadratic regularizer. For such a model, we can solve Eq. 5 with gradient descent (GD),

$$\boldsymbol{\theta}_{t+1} = (1 - \rho_t)\boldsymbol{\theta}_t - \rho_t \sum_{i=1}^{N} \boldsymbol{\phi}_i \left[\sigma(f_i(\boldsymbol{\theta}_t)) - y_i\right] \quad (8)$$

The result is obtained by simply taking the derivative of Eq. 7 which gives rise to $\sigma(f) = 1/(1 + e^{-f})$, a binary version of the softmax function from Eq. 4. We will now show that, by choosing the family $\mathcal{Q}$ appropriately, variational learning can be seen as GD with label noise.

We choose the distribution $q_t(\boldsymbol{\theta})$ at iteration $t$ to take a Gaussian form with mean $\boldsymbol{\theta}_t$ and covariance set to the identity,

$$q_t(\boldsymbol{\theta}) = \mathcal{N}(\boldsymbol{\theta}|\boldsymbol{\theta}_t, \mathbf{I}),$$

and perform GD to minimize the variational objective in Eq. 6, now denoted as $\mathcal{L}(\boldsymbol{\theta}_t)$, with respect to $\boldsymbol{\theta}_t$. Below is a formal statement of the result.

**Theorem 1** *A gradient update $\boldsymbol{\theta}_{t+t} = \boldsymbol{\theta}_t - \rho_t \nabla_{\boldsymbol{\theta}_t} \mathcal{L}(\boldsymbol{\theta}_t)$ is equivalent to the gradient update in Eq. 8 where the label $y_i$ are replaced by $y_i + \epsilon_{i|t}$ with noise defined as*

$$\epsilon_{i|t} = \sigma(f_i(\boldsymbol{\theta}_t)) - \mathbb{E}_{q_t}[\sigma(f_i(\boldsymbol{\theta}))]. \quad (9)$$

**Proof:** The gradient of the expected loss in Eq. 6 can be simplified to take a form very similar to the one in Eq. 8,

$$\begin{aligned}
\nabla_{\boldsymbol{\theta}_t} \mathbb{E}_{q_t}[\ell_i(\boldsymbol{\theta})] &= \nabla_{\boldsymbol{\theta}_t} \mathbb{E}_{\mathcal{N}(\mathbf{e}|0,\mathbf{I})}[\ell_i(\boldsymbol{\theta}_t + \mathbf{e})] \\
&= \mathbb{E}_{\mathcal{N}(\mathbf{e}|0,\mathbf{I})}[\nabla_{\boldsymbol{\theta}_t} \ell_i(\boldsymbol{\theta}_t + \mathbf{e})] \quad (10) \\
&= \boldsymbol{\phi}_i \left[\mathbb{E}_{q_t}[\sigma(f_i(\boldsymbol{\theta}))] - y_i\right]
\end{aligned}$$

The gradient of KL is also simplifies to

$$\mathbb{D}_{\mathrm{KL}}[q(\boldsymbol{\theta}) \,\|\, p(\boldsymbol{\theta})] = \mathbb{E}_q\left[\log \frac{q(\boldsymbol{\theta})}{p(\boldsymbol{\theta})}\right] = \frac{1}{2}\|\boldsymbol{\theta}_t\|^2 + \text{const.}$$

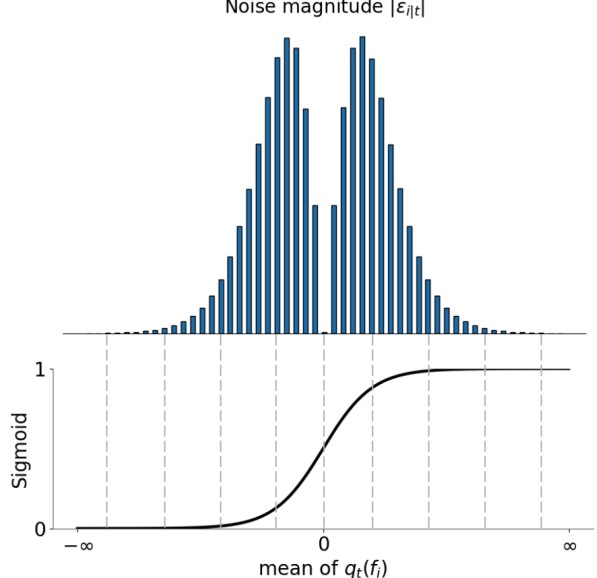

Figure 2. We plot label noise magnitude $\epsilon_{i|t}$ from Eq. 12 by varying the mean $f_{i|t}$ of $q_t(f_i)$ while fixing its variance to 1. The noise is large around 0 (but not at 0) with large peaks on both sides.

Using these, we can write the GD to minimize Eq. 6 as

$$\boldsymbol{\theta}_{t+1} = (1 - \rho_t)\boldsymbol{\theta}_t - \rho_t \sum_{i=1}^{N} \boldsymbol{\phi}_i \left[\mathbb{E}_{q_t}[\sigma(f_i(\boldsymbol{\theta}))] - y_i\right],$$

which has a similar form Eq. 8 but with one difference: $\sigma(f_i(\boldsymbol{\theta}))$ are replaced by their expectation over $q$ (highlighted in red). By adding and subtracting $\sigma(f_i(\boldsymbol{\theta}_t))$, we can rewrite the update as Eq. 8 which has the label noise defined in Eq. 9. ∎

The result shows that the GD steps to optimize Eq. 6 is equivalent to those to optimize Eq. 5 but with a noisy label. The noise is adaptive and depends on where the Gaussian distribution is located. To show this, we derive the distribution over $f_i = \boldsymbol{\phi}_i^\top \boldsymbol{\theta}$, which takes a Gaussian form:

$$q_t(f_i) = \mathcal{N}(f_i|f_{i|t}, \boldsymbol{\phi}_i^\top \boldsymbol{\phi}_i), \quad (11)$$

where we denote $f_{i|t} = \boldsymbol{\phi}_i^\top \boldsymbol{\theta}_t$. The label noise then is simply the difference between the sigmoid $\sigma(f_{i|t})$ of the mean $f_{i|t}$ and mean of $\sigma(f_i)$ with respect to $q_t(f_i)$, that is

$$\epsilon_{i|t} = \sigma(f_{i|t}) - \mathbb{E}_{q_t}[\sigma(f_i)]. \quad (12)$$

Fig. 2 plots the magnitude of this quantity as a function of the mean $f_{i|t}$ but fixing the variance $\boldsymbol{\phi}_i^\top \boldsymbol{\phi}_i = 1$. We see the noise to be large whenever $f_{i|t}$ around 0, with the maximum in areas slightly away from it. The $\sigma(f)$ is flat far away from 0 and uncertainty in $q_t$ is amplified around 0, which makes the difference also large away from 0 (but not at 0).

The other factor that affects the noise is the feature $\phi_i$. Inputs with larger features induce larger variance. When the features are normalized, this is unlikely to have an effect, but this is important for the neural networks case where features are learned.

The two factors explain why we would expect high label noise for atypical or ambiguous examples. This is because the predictive distribution $q(f_{i|t})$ is close to 0 and may also have a higher variance. An alternate way to understand the impact of the two factors is to use a Taylor's approximation at a sample $e \sim \mathcal{N}(0, 1)$,

$$\epsilon_{i|t} \approx \sigma(f_{i|t}) - \sigma(f_{i|t} + \|\phi_i\|_2 e) \approx \sigma'(f_{i|t})(\phi_i^\top \phi_i)^{1/2} e. \tag{13}$$

We again see the two factors: one is $\sigma'(f)$ (which peaks around 0) and the other is the feature norm. Note that this approximation does not get better for larger number of samples, but it roughly captures the behavior away from 0.

### 3.2. Generalized Linear Model (GLM) with GD

The result generalizes to any loss function derived using exponential-family distribution, for instance, the following generalization of Eq. 7

$$\ell_i(\boldsymbol{\theta}) = -\mathbf{y}_i^\top \mathbf{f}_i(\boldsymbol{\theta}) + A(\mathbf{f}_i(\boldsymbol{\theta})), \tag{14}$$

where $A(\mathbf{f})$ is a convex function called the log-partition function. The regularize can also be a general convex function. For such models, we can derive the label noise following almost the same procedure as in the previous section. Due to its similarity, we omit the derivation and only give the final form of the noise,

$$\boldsymbol{\epsilon}_{i|t} = A'(\mathbf{f}_i(\boldsymbol{\theta}_t)) - \mathbb{E}_{q_t}[A'(\mathbf{f}_i(\boldsymbol{\theta}))]. \tag{15}$$

Essentially, we replace the $\sigma(f)$ by the derivative $A'(f)$. For logistic regression, $A(f) = \log(1 + e^f)$, derivative of which is $\sigma(f)$ and we recover the result in Eq. 9. We can extend this result to multiclass classification by considering $A(\mathbf{f}) = \log \sum_{k=1}^{K} e^{f_k}$, derivative of which is the softmax function defined in Eq. 4. Similarly to the binary case, we expect uncertainty in $q_t$ to be amplified near the boundary. The label noise is therefore low for examples where softmax yields probabilities close to 0 or 1.

### 3.3. Generalized Linear Model with Newton's Method

We now go beyond GD to Newton's method and show that a specific variational-learning algorithm can be seen as a noisy-label version of Newton's method. This is a useful step before we move to neural networks training. Here, we find that the form of the noise has exactly same form as Eq. 15 but the distribution $q_t$ has flexible covariance which improves the adaptivity of the label noise.

We consider the following Newton's update,

$$\boldsymbol{\theta}_{t+1} = \boldsymbol{\theta}_t - \left[\nabla^2 \bar{\ell}(\boldsymbol{\theta}_t)\right]^{-1} \nabla \bar{\ell}(\boldsymbol{\theta}_t) \tag{16}$$

which is commonly used for generalized linear models. As shown by Khan & Rue (2023), the update can be seen as a special case of a Variational Online Newton (VON) algorithm to learn a full Gaussian with covariance $\boldsymbol{\Sigma}_t$,

$$q_t(\boldsymbol{\theta}) = \mathcal{N}(\boldsymbol{\theta}|\boldsymbol{\theta}_t, \boldsymbol{\Sigma}_t)$$

The VON updates are given as follows,

$$\begin{aligned} \boldsymbol{\theta}_{t+1} &= \boldsymbol{\theta}_t - \rho_t \boldsymbol{\Sigma}_{t+1} \mathbb{E}_{q_t}[\nabla \bar{\ell}(\boldsymbol{\theta})] \\ \boldsymbol{\Sigma}_{t+1}^{-1} &= (1 - \rho_t)\boldsymbol{\Sigma}_t^{-1} + \rho_t \mathbb{E}_{q_t}[\nabla^2 \bar{\ell}(\boldsymbol{\theta})]. \end{aligned} \tag{17}$$

Setting $\rho_t = 1$ yields a Newton-like update where gradients $\nabla \bar{\ell}$ and Hessian $\nabla^2 \bar{\ell}$ are replaced by their terms where expectations are taken, namely, $\mathbb{E}_{q_t}[\nabla \bar{\ell}]$ and $\mathbb{E}_{q_t}[\nabla^2 \bar{\ell}]$. Similarly to the previous cases, the label noise in VON arises due to the expectation of the gradient, while expectation of the Hessian gives rise to other types of noise.

As shown in in App. A.1, the VON updates in Eq. 17 are equivalent to Newton's update in Eq. 16 where labels are replaced by the noisy ones with noise shown in Eq. 15. The proof technique relies on comparing the form of the surrogates for the two algorithms. Even though the noise has the same form, there is an important difference here. Essentially, the Gaussian $q_t$ now is more flexible because its covariance $\boldsymbol{\Sigma}_t$ is not fixed but learned using the Hessian. As a result the distribution over $f_i$ now has adaptive variances,

$$q_t(f_i) = \mathcal{N}(f_i|f_{i|t}, \phi_i^\top \boldsymbol{\Sigma}_t \phi_i), \tag{18}$$

Therefore, now both the location and spread of the Gaussians are changed for each example, and they both contribute to the adaptivity. The result shows that second-order methods yield more adaptive label noise than first order methods, and are expected to perform better in practice. We will later present experiments that support this finding.

### 3.4. Neural Network training with IVON

We will now show that the label noise expression have similar form for the neural network case, but to derive them we need to use Taylor's approximation. Essentially, the form of the expression then is similar to Eq. 13 there the adaptive nature should roughly stay the same. We validate these findings later through numerical experiments.

We will illustrate the derivation for the binary case which can then be extended to other case as we did in previous section. Taylor's approximations is required because the gradient of $\ell_i$, shown below,

$$\nabla \ell_i(\boldsymbol{\theta}) = \nabla f_i(\boldsymbol{\theta}_t) \left[\sigma(f_i(\boldsymbol{\theta}_t)) - y_i\right],$$

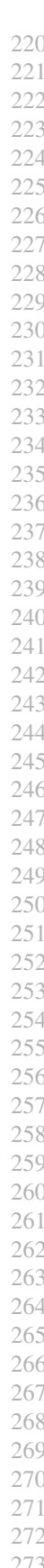
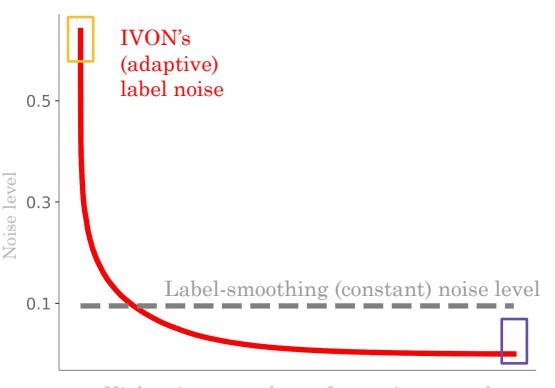
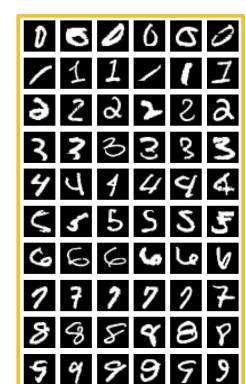
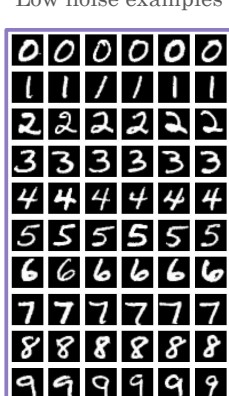

*Figure 3.* Label noise assigned by IVON and LS in MNIST dataset. Examples are ordered according to IVON's noise, and highest and lowest noise examples are visualized. We see that high noise is assigned to atypical examples while low noise is assigned to regular ones.

replaces the $\phi_i$ term in Eq. 8 by $\nabla f_i(\boldsymbol{\theta}_t)$. As a result, we cannot simply move the expectation over $q_t$ to derive the label noise as we did in Eq. 10. However, we can simplify these by using Taylor's approximation.

We show this by using a single-sample $\boldsymbol{\theta}_t^{(1)} \sim q_t$ Monte-Carlo approximation (multiple samples can also be used),

$$\mathbb{E}_{q_t}\left[\nabla \ell_i(\boldsymbol{\theta})\right] \approx \nabla f_i(\boldsymbol{\theta}_t^{(1)}) \left[\sigma(f_i(\boldsymbol{\theta}_t^{(1)})) - y_i\right].$$

Then, we do the following two approximations where we use Taylor's expansion but ignore the second-order terms,

$$\sigma(f_i(\boldsymbol{\theta}^{(1)})) \approx \sigma(f_i(\boldsymbol{\theta}_t)) + \sigma'(f_i(\boldsymbol{\theta}_t))\nabla f_i(\boldsymbol{\theta}_t)(\boldsymbol{\theta}^{(1)} - \boldsymbol{\theta}_t)$$
$$\nabla f_i(\boldsymbol{\theta}^{(1)}) \approx \nabla f_i(\boldsymbol{\theta}_t)$$

With these approximations, we can write,

$$\mathbb{E}_{q_t}\left[\nabla \ell_i(\boldsymbol{\theta})\right] \approx \nabla f_i(\boldsymbol{\theta}_t) \left[\sigma(f_i(\boldsymbol{\theta}_t)) - (y_i + \epsilon_{i|t})\right]$$

where the noise takes a very similar form to Eq. 13

$$\epsilon_{i|t} \approx \sigma'(f_i(\boldsymbol{\theta}_t))\nabla f_i(\boldsymbol{\theta}_t)\boldsymbol{\Sigma}_t^{1/2}\mathbf{e} \qquad (19)$$

where $\mathbf{e}$ is a sample from a standard normal distribution. The derivation generalizes to all GLM losses by replacing $\sigma(\cdot)$ by $A'(\cdot)$. It also extends to variational GD and VON.

In practice, neural networks are trained with Adam-style algorithm. In our experiments, we will use an Adam-like version of VON, called IVON, which is recently proposed by Shen et al. (2024). The key different to VON is that it estimates a diagonal covariance by using an Adam-like preconditioning update; a pseudo-code is added in Alg. 1. The diagonal covariance is estimated through the scale vector. We will use the label noise expression given in Eq. 19 where $\boldsymbol{\Sigma}_t$ is replaced by the diagonal covariance estimated

by IVON. Note that variational learning for neural neworks with IVON introduces many other noise other than label noise, for instance, the noise is introduced in the features $\nabla f(\boldsymbol{\theta})$, as shown above. We will analyze only the label noise but the performance is affected by other noises too.

In our experiments, we also compare to Sharpness-Aware Minimization (SAM) (Foret et al., 2020) which has a variational interpretation (Möllenhoff & Khan, 2022) and has been shown to perform well with mislabelled data. Using our techniques, it is possible to derive the label noise of SAM but the expression would be similar to the one derived here. The difficulty with SAM is that we need to tune the 'size' of adversarial perturbation, often denoted by a scalar $\rho$, IVON can automatically estimate it using the posterior variance. In our experiments, we show that IVON performs comparably to SAM with a highly tuned $\rho$, and it does not need to set any such hyperparameters.

## 4. Experiments

We do extensive experiments to show adaptive label noise via variational learning and its benefits. In Sec. 4.1, we show that IVON adapts the label noise for each examples, and generally assigning higher noise magnitude to ambiguous ones. In Sec. 4.2, we show that IVON's smoothed labels are similar to an existed adaptive smoothing method (Zhang et al., 2021). In Sec. 4.3, we show that IVON consistently outperforms LS when datasets have labeling errors in various settings. Additional experiments are reported in App. B, and experiment details are reported in App. C.

### 4.1. IVON's Adaptive Label Noise

We demonstrate IVON label noise's adaptivity on MNIST dataset (LeCun & Cortes, 2010). We plot IVON's label

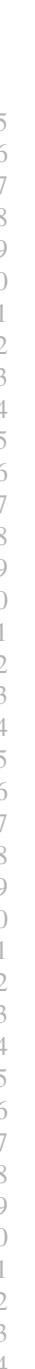
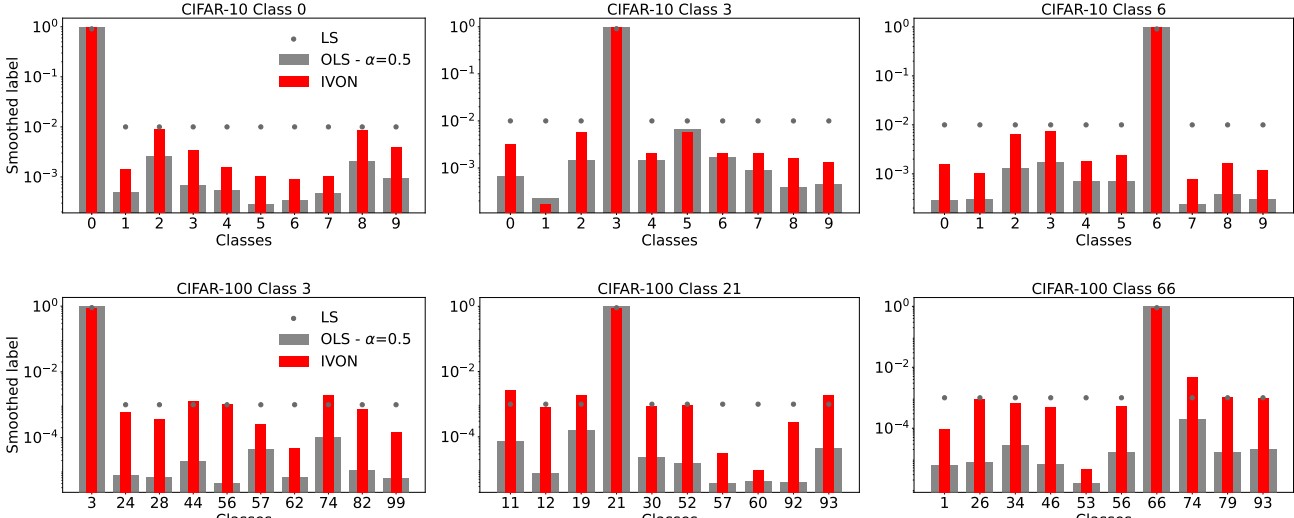

*Figure 4.* Smoothed label comparison among IVON (Shen et al., 2024), LS (Szegedy et al., 2016) and Online Label Smoothing (OLS) (Zhang et al., 2021). IVON has similar adaptive label smoothing effect as OLS. $\alpha$ is the smoothing rate defined in Eq. 1. Y-axis is in the log scale. We randomly pick 10 classes for CIFAR-100 due to image size limit.

noise distribution in Fig. 3, which shows that IVON adds different label noise on each example whereas traditional Label Smoothing defines a uniformly distributed noise for all. By further visualizing the data, we see that IVON induces stronger noise to unclear examples, which prevent models from being overconfident on these datapoints.

### 4.2. Comparisons to Existing Adaptive LS Strategies

In this section, we show that IVON's label smoothing is similar to an adaptive method called Online Label Smoothing (OLS) (Zhang et al., 2021). In the CIFAR-10 and CIFAR-100 dataset (Krizhevsky & Hinton, 2009), we compare the smoothed labels of IVON with traditional LS (Szegedy et al., 2016) and Online Label Smoothing (OLS) (Zhang et al., 2021). OLS adjusts the label noise according to the model's predictions, as described in Sec. 2. As Fig. 4 shows, IVON has surprisingly similar smoothed label distributions as the OLS in both datasets, while IVON tends to stronger label noises. Variational learning's adaptive label smoothing is similar to existing work's, without needing any additional effort to design or estimate the adaptive label noise.

### 4.3. Comparisons on Datasets with Labeling Errors

We compare IVON to Label Smoothing (LS) (Szegedy et al., 2016) and SAM (Foret et al., 2020) in presence of labeling errors, and the results show that IVON consistently outperforms LS in various settings. To find the best performance of the baselines, we tune several LS's smoothing rates $\alpha$ (defined in Eq. 1), and various SAM's adversarial perturbation size $\rho$ (discussed in Sec. 3.4). We conduct studies on bench-

mark datasets with synthetic noise, where the noise level can be adjusted, followed by evaluations on datasets with natural noise, where the noise level is fixed and unknown. For synthetic noise experiments, we use the CIFAR-10 and CIFAR-100 datasets (Yu et al., 2019). For natural noise experiments, we use the benchmark Clothing1M (Xiao et al., 2015). All datasets include a clean test set.

#### 4.3.1. SYNTHETIC NOISY DATASETS

We consider two commonly used corruptions (Patrini et al., 2017; Li et al., 2019; Yu et al., 2019): Symmetric flipping and Pair flipping. In symmetric flipping, a true label is replaced by a randomly generated class with a probability. In pair flipping, it tries to mimic real world mistakes for similar classes, where a true label is replaced by the next class with a probability. For training dataset, we use previous work's (Yu et al., 2019) code to generate noisy labels. More experiment details are in App. C.2.

In CIFAR-10, Fig. 5 shows that IVON outperforms Label Smoothing and SAM in different scenarios. We also observe that SAM is sensitive to the choice of $\rho$, while IVON does not need to tune any hyperparameters to perform well. In CIFAR-100, Fig. 8 shows similar trends. For instance, in pairflip 20% noise setting, IVON outperforms best LS performance by 9.1% and best SAM performance by 13.3%.

Meanwhile, we test the effectiveness of flexible $\mathbf{\Sigma}_t$ by comparing it with the fixed diagonal covariance. Fig. 6 shows that learned $\mathbf{\Sigma}_t$ consistently outperforms fixed $\mathbf{\Sigma}_t$ in three noisy datasets. The experiment results demonstrate the importance of flexible $\mathbf{\Sigma}_t$ as stated in Sec. 3.3.

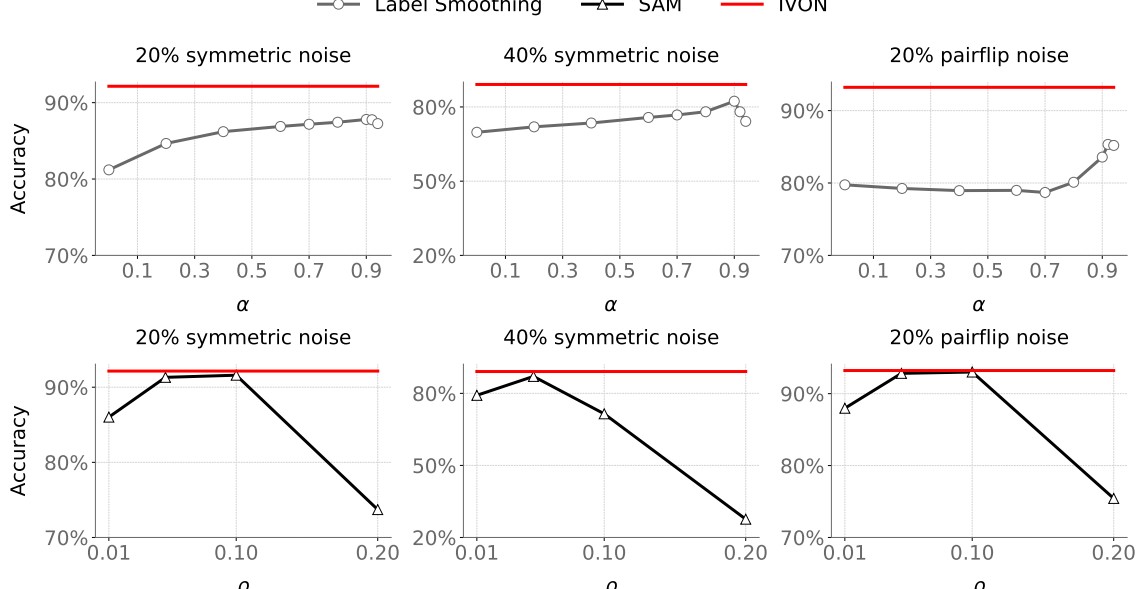

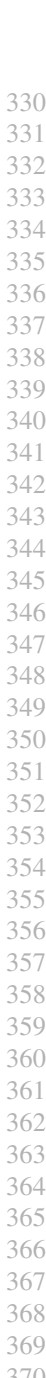

*Figure 5.* Results on CIFAR-10 with symmetric noisy labels. Top: IVON outperforms Label Smoothing (LS) with different smoothing rates $\alpha$. When comparing with LS peak performances, IVON outperforms by 4.3%, 6.7% and 7.8% in three datasets, from left to right, respectively. Down: IVON has comparable results with SAM peak performances, while SAM is sensitive to the choice of perturbation $\rho$. Results are reported over 5 random seeds.

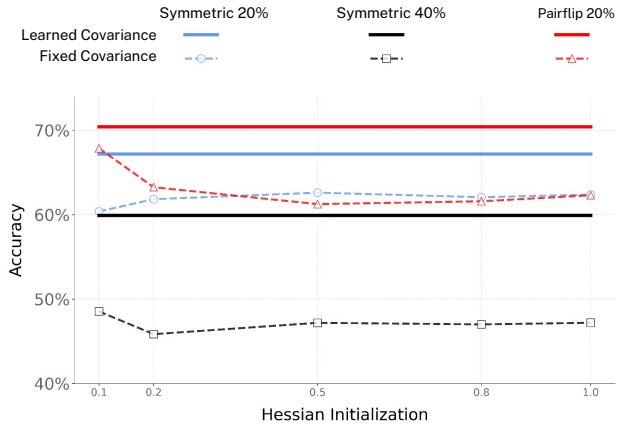

*Figure 6.* In synthetic noisy datasets of CIFAR-100, we test IVON with multiple fix Hessian, which fix the diagonal covariance $\Sigma_t$ of weight posterior as described in App. C.2. The fixed diagonal $\Sigma_t$ is worse than learned diagonal covariance in all datasets.

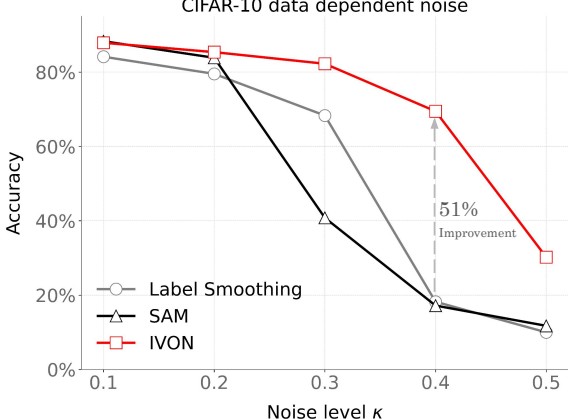

*Figure 7.* Results for CIFAR-10 with data dependent noise. IVON consistently outperforms LS and SAM in all noise levels. Furthermore, IVON can learn extremely noisy scenario, while LS and SAM cannot.

### 4.3.2. DATA DEPENDENT LABELING ERRORS

In this experiment, we try to understand the adaptivity of these methods in data-dependent noisy dataset. When each class has different noise levels, we expect LS will fail since it induce uniform noises to all classes, while IVON's adaptivity makes it stand.

First, we create a new transition matrix $P$ of noisy label

$\mathbf{y}' = P\mathbf{y}$, where $\mathbf{y}, \mathbf{y}' \in \mathbb{R}^K, P \in \mathbb{R}^{K \times K}$. We inject difference noise level to each class, so the noise level of each class is different:

$$P_{i,i} = 1 - (\kappa + \beta i), i \in [1, K]. \quad (20)$$

where $\kappa$ is the starting noise level and $\beta$ is the increase factor. Afterwards, we give the same transition probability

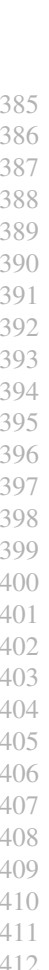
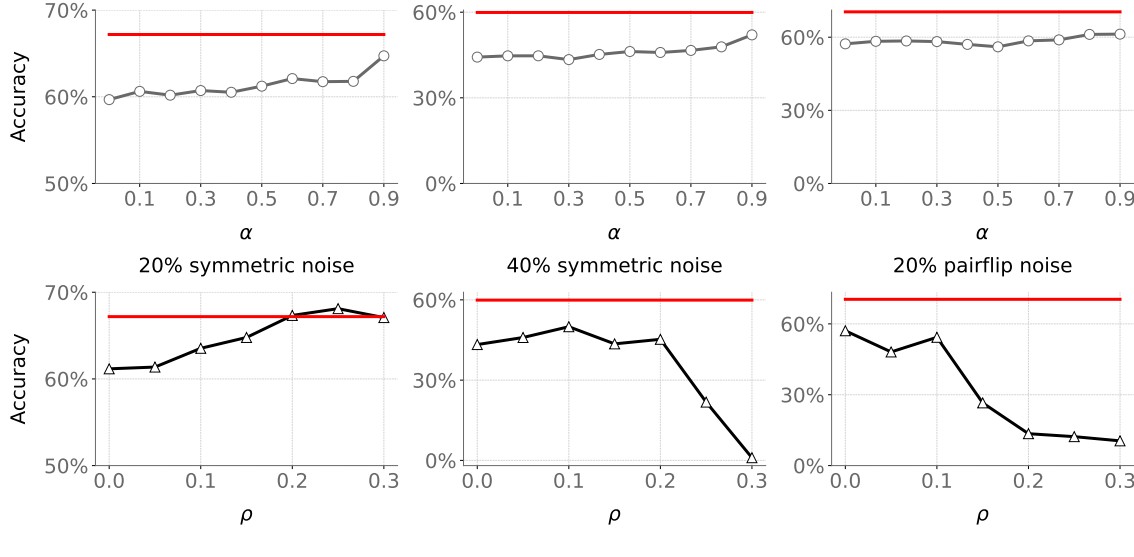

*Figure 8.* Results on CIFAR-100 with symmetric noisy labels over 5 random seeds, which is similar to CIFAR-10 results in Fig. 5. IVON outperforms LS by 2.4%, 7.8% and 9.1% in three datasets respectively. Meanwhile, IVON outperforms SAM by 9.9% in 40% symmetric noise dataset and 13.3% in 20% pairflip noise dataset.

to the rest of the wrong classes:

$$P_{i,j} = \frac{\kappa + \beta i}{K - 1}, i, j \in [1, K], i \neq j. \tag{21}$$

In experiments, we follow the hyperparameters in CIFAR-10 synthetic noise experiment from Sec. 4.3.1. For LS, we run smoothing rate $\{0, 0.1, 0.3, 0.5, 0.7, 0.9\}$ and report the best accuracy. For SAM, we run $\rho$ for $\{0.0, 0.05, 0.1, 0.15, 0.2, 0.5\}$ and report the best accuracy of them.

The experiment results for $\kappa = \{0.1 \sim 0.5\}$ and $\beta = 0.05$ are in Fig. 7. Overall, IVON outperforms LS and SAM in all noise levels. Meanwhile, IVON can learn in very noisy scenarios $\kappa = \{0.4, 0.5\}$ while baselines can only reach around 10% accuracy. The experiment results support our claim that adaptive label noise induced by variational learning is more effective than traditional label smoothing.

### 4.3.3. UNCONTROLLED NOISY DATASETS

We now report results on Clothing1M (Xiao et al., 2015), a large-scale dataset that features natural label noise from the web and consists of 1 million images across 14 categories. We conduct experiments by using ResNet-50 as the model.

The results on Clothing1M, illustrated in Fig. 9, demonstrate that IVON outperforms Label Smoothing and is comparable to SAM. This experiment shows that IVON's performance is consistent in the large scale dataset.

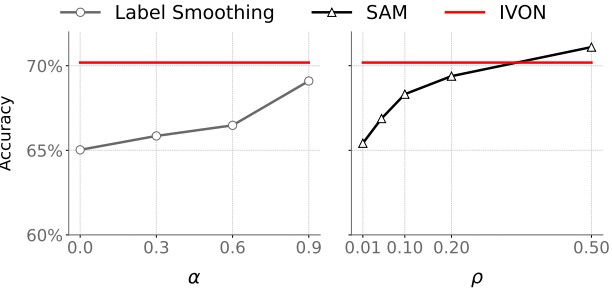

*Figure 9.* Clothing 1M experiment result. The result is similar to synthetic noisy datasets reported in Fig. 5 and Fig. 8. Results are reported over 5 seeds.

## 5. Conclusion

In this paper, we show that variational learning induces adaptive label smoothing. We show such adaptive label noise naturally emerges in variational learning without additional mechanisms. We derive the exact form of the label noise, and do extensive experiments to show its benefits. Empirical results demonstrate that variational learning assigns stronger label noise to ambiguous examples, induces similar noise distributions as an existing adaptive method does (Zhang et al., 2021), and consistently outperforms label smoothing in the presence of labeling errors. From the Bayesian methods perspective, we present a new way to introduce label smoothing.

## Impact Statement

This paper presents work whose goal is to advance the field of Machine Learning. There are many potential societal consequences of our work, none which we feel must be specifically highlighted here.

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

## A. Derivations

### A.1. Derivation of GLM with Newton's method

For a Newton's update, it is equivalent to the following surrogate minimization,

$$\boldsymbol{\theta}_{t+1} = \arg \min_{\boldsymbol{\theta}} \left[ \boldsymbol{\theta}^\top \nabla \bar{\ell}(\boldsymbol{\theta}_t) + \frac{1}{2}(\boldsymbol{\theta} - \boldsymbol{\theta}_t)^\top \nabla^2 \bar{\ell}(\boldsymbol{\theta}_t)(\boldsymbol{\theta} - \boldsymbol{\theta}_t) \right]. \tag{22}$$

We will show the noisy version of such Newton's update are implicitly induced by the Bayesian learning rule to estimate a full Gaussian $q_t(\boldsymbol{\theta}) = \mathcal{N}(\boldsymbol{\theta}|\boldsymbol{\theta}_t, \boldsymbol{\Sigma}_t)$ with full covariance $\boldsymbol{\Sigma}_t$,

$$q_{t+1}(\boldsymbol{\theta}) \propto q_t(\boldsymbol{\theta})^{1-\rho_t} \prod_{i=1}^{N} e^{\rho_t \left( \boldsymbol{\theta}^\top \mathbb{E}_{q_t}[-\nabla \ell_i(\boldsymbol{\theta}) + \nabla^2 \ell_i(\boldsymbol{\theta}) \cdot \boldsymbol{\theta}_t] - \frac{1}{2} \boldsymbol{\theta}^\top \mathbb{E}_{q_t}[\nabla^2 \ell_i(\boldsymbol{\theta})] \boldsymbol{\theta} \right)}, \tag{23}$$

as shown in Nickl et al. (2024) App. C.3. By comparing these two equations, we can show the following result.

**Theorem 2** *For the loss function in Eq. 14, the Bayesian learning rule shown in Eq. 23 is equivalent to Newton's update in Eq. 22 with a noisy label $\ell(y_i + \epsilon_{i|t}, \boldsymbol{\theta})$. The noise depends on the variance $\boldsymbol{\phi}_i^\top \boldsymbol{\Sigma}_t \boldsymbol{\phi}_i$, as shown below:*

$$\epsilon_{i|t} = A'\left( f_{i|t} \right) - \mathbb{E}_{\mathcal{N}(e|0, \boldsymbol{\phi}_i^\top \boldsymbol{\Sigma}_t \boldsymbol{\phi}_i)}\left[ A'\left( f_{i|t} + e \right) \right]. \tag{24}$$

*There is additional noise introduced in the Hessian which depends on $A''(f_{i|t})$.*

### A.2. IVON pseudo code

---

**Algorithm 1** Improved Variational Online Newton (IVON) (Shen et al., 2024).

---

**Require:** Learning rates $\{\alpha_t\}$, weight-decay $\delta > 0$.
**Require:** Momentum parameters $\beta_1, \beta_2 \in [0, 1)$.
**Require:** Hessian init $h_0 > 0$.
**Init:** $\mathbf{m} \leftarrow$ (NN-weights), $\mathbf{h} \leftarrow h_0$, $\mathbf{g} \leftarrow 0$, $\lambda \leftarrow N$.
**Init:** $\boldsymbol{\sigma} \leftarrow 1/\sqrt{\lambda(\mathbf{h} + \delta)}$.
**Optional:** $\alpha_t \leftarrow (h_0 + \delta)\alpha_t$ for all $t$.
 1: **for** $t = 1, 2, \ldots$ **do**
 2: $\quad \widehat{\mathbf{g}} \leftarrow \widehat{\nabla} \bar{\ell}(\boldsymbol{\theta})$, where $\boldsymbol{\theta} \sim q$
 3: $\quad \widehat{\mathbf{h}} \leftarrow \widehat{\mathbf{g}} \cdot (\boldsymbol{\theta} - \mathbf{m})/\boldsymbol{\sigma}^2$
 4: $\quad \mathbf{g} \leftarrow \beta_1 \mathbf{g} + (1 - \beta_1)\widehat{\mathbf{g}}$
 5: $\quad \mathbf{h} \leftarrow \beta_2 \mathbf{h} + (1 - \beta_2)\widehat{\mathbf{h}} + \frac{1}{2}(1 - \beta_2)^2(\mathbf{h} - \widehat{\mathbf{h}})^2/(\mathbf{h} + \delta)$
 6: $\quad \bar{\mathbf{g}} \leftarrow \mathbf{g}/(1 - \beta_1^t)$
 7: $\quad \mathbf{m} \leftarrow \mathbf{m} - \alpha_t(\bar{\mathbf{g}} + \delta \mathbf{m})/(\mathbf{h} + \delta)$
 8: $\quad \boldsymbol{\sigma} \leftarrow 1/\sqrt{\lambda(\mathbf{h} + \delta)}$
 9: **end for**
10: **return** $\mathbf{m}, \boldsymbol{\sigma}$

---

## B. Additional Experiments

### B.1. Hessian Initialization

We analyze how Hessian initialization $h_0$ of IVON affects the accuracy. The results are in Fig. 10. IVON's accuracy can only vary by up to 10% when the Hessian is bigger than 0.05, and this variation is less sensitive compared to SAM's sensitivity to $\rho$, as shown in Fig. 5, Fig. 8 and Fig. 9.

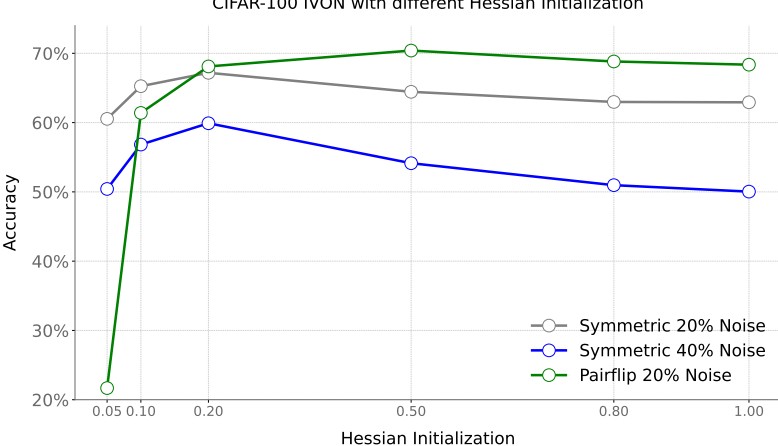

*Figure 10.* Results for IVON on CIFAR-100 with multiple Hessian initialization. IVON's accuracy is consistent when having different Hessian initializations.

## C. Experiment details

### C.1. Experiment details of Sec. 4.1 and Sec. 4.2

In Fig. 3, we test IVON on a 3-layers convolutional neural networks. In Fig. 4, we do experiments on ResNet-34 model. We uses the PyTorch implementation verison[1] of Online Label Smoothing (Zhang et al., 2021).

### C.2. Experiments on Synthetic Noisy Datasets

For pairflip setting in CIFAR-10, the classes flipping order is: AIRPLANE → AUTOMOBILE → BIRD → CAT → DEER → DOG → FROG → HORSE → SHIP → TRUCK → AIRPLANE. In CIFAR-10 experiments, we train a ResNet 34 for 200 epochs with batch size set to 50 and weight decay set to 0.001. For SAM and LS, we set initial learning rate as 0.05 and reduce it by 0.1 at 100 epoch and 150 epoch, following hyper-parameters from previous papers. For IVON (Shen et al., 2024), we follow the original paper to set initial learning rate as 0.2 and anneal the learning rate to zero with a cosine schedule after a linear warmup phase over 5 epochs. We set momentum to 0.9 for all methods, and hessian momentum $\beta_2$ to $1 - e^{-5}$, hessian initial $h_0$ to 0.9, scaling parameter $\lambda$ to the number of training data for IVON. For SAM, we follow the original paper (Foret et al., 2020) and choose best neighborhood size $\rho$ from $[0.01, 0.05, 0.1, 0.2, 0.5]$. In CIFAR-100 experiments, we tune the hyperparamters to the best for each method. The hyperparameters are specified in Table 1.

In Fig. 6, we fix the Hessian of IVON by setting $\beta_2 = 1$ in Line 5 of Alg. 1. Therefore, covariance $\boldsymbol{\sigma}$ defined in Line 8 is fixed since Hessian $\mathbf{h}$ is fixed.

*Table 1.* Hyperparamters of each method for CIFAR-100. We denote learning rate as lr, Hessian Initialization as Hessian init.

|  | Symmetric 20% | Symmetric 40% | Pairflip 20% |
|---|---|---|---|
| Weight decay | 2e-4 | 2e-4 | 5e-4 |
| LS (Szegedy et al., 2016) lr | 0.1 | 0.1 | 0.1 |
| SAM (Foret et al., 2020) lr | 0.05 | 0.1 | 0.1 |
| IVON (Shen et al., 2024) lr | 0.8 | 0.8 | 0.5 |
| IVON (Shen et al., 2024) Hessian init | 0.2 | 0.2 | 0.5 |

### C.3. Clothing 1M Details

The noisy labels in Clothing1M (Xiao et al., 2015) are derived from the text surrounding the images on the web. In constructing the dataset, noisy labels are assigned to images based on this contextual text.

---

[1]https://github.com/ankandrew/online-label-smoothing-pt

