# OpenReview forum: "Variational Learning Induces Adaptive Label Smoothing"
_ICML.cc/2025/Conference — Submitted to ICML 2025_

### Official Review · Reviewer_KjjK · 2025-03-07

**Overall Recommendation:** 2

**Summary:**

The author finds that variational learning naturally induces an adaptive label smoothing where label noise is specialized for each example. As a result, variational learning method IVON shows performance gain compared with traditional label smooth method.

## update after rebuttal

My concerns are not well addressed by the rebuttal, and I maintain the score.

**Claims And Evidence:**

I think the conclusions in Section 3.4 lack credibility. Reducing the expectation merely to the sampling of a single instance is unreasonable. The authors should provide results for multiple samples. Additionally, while not mandatory, it would be preferable for the authors to apply some statistical methods to provide corresponding error estimates.

**Essential References Not Discussed:**

No problem.

**Experimental Designs Or Analyses:**

No problem.

**Methods And Evaluation Criteria:**

If the author’s claim can be really convinced, it’s useful for label smoothing field.

**Other Comments Or Suggestions:**

Since Section 3.2 is a generalization of Section 3.1 and both follow the same approach, why not provide a detailed proof for Section 3.2 while treating Section 3.1 as a special case? It is uncommon to present a detailed proof for the simpler case while skimming over the more general one.

**Other Strengths And Weaknesses:**

The major issue of this paper lies in its writing logic. Although the author's proofs are easy to understand, the writing is confusing, making it difficult for me to comprehend many parts. I have listed these confusions in the questions below. Before clarifying these confusions, I prefer not to hastily categorize some issues as weaknesses.

**Questions For Authors:**

- I did not understand the discussion after Thm.1 (Lines 145–183). Shouldn't the target predicted by $f_i$ be the 0/1 labels? Why does a symmetric situation arise? Please provide a detailed explanation, as I do not understand the purpose, conclusion, or the meaning conveyed by Figure 2 in this section.
- What is the role of Section 3.3? How does it help in deriving the conclusions of Section 3.4?
- The approach in Section 3.4 seems overly simplistic. Is it truly reasonable?
- The proof appears to heavily rely on the linear term $-yf(x)$ in the loss function. Does the result hold for other loss functions as well? Can the class of loss functions discussed by the authors encompass common loss functions such as cross-entropy loss? What effect do they have on different losses (such as different formats in exponential-family loss and some other loss functions beyond exponential-family loss)?

**Relation To Broader Scientific Literature:**

Showing that variational learning induces an adaptive label smoothing.

**Theoretical Claims:**

There’s no problem in Thm.1. While for Thm.2, the author does not give a proof, which I think should be added.

---

> ### Author Rebuttal · Authors · 2025-04-01
>
> We thank the reviewer for their review. The main issue seems to be related to the writing of the paper, for which we have given clarifications and promised a few fixes. We hope that the reviewer will consider raising their score.
>
> > Q1: I think the conclusions in Section 3.4 lack credibility. Reducing the expectation merely to the sampling of a single instance is unreasonable. The authors should provide results for multiple samples.
>
> A1: There is a misunderstanding. In Sec 3.4, we show a single sample for simplicity but state that multiple samples are also possible (see line 244). In fact, in our experiments, we do use multiple samples. Hope this clarifies that there is no issue with credibility.
>
> > Q2 It would be preferable for the authors to apply some statistical methods to provide corresponding error estimates.
>
> A2: Could the reviewer clarify what they mean by “error estimates”?
>
> > Q3: While for Thm.2, the author does not give a proof, which I think should be added.
>
> A3: Thanks. We will expand the proof and make it easier to understand.
>
> > Q4: The issue of this paper lies in its writing logic.. The writing is confusing…Since Section 3.2 is a generalization of Section 3.1 and both follow the same approach, why not provide a detailed proof for Section 3.2 while treating Section 3.1 as a special case?
>
> A4: It is an interesting suggestion, we will try to revise it with your suggestions.
>
> > Q5: Shouldn’t the target predicted by fi be the 0/1 labels?
>
> A5: No. The $f_i \in \mathbb{R}$ takes real values, while the sigmoid $\sigma(f_i) \in [0,1]$ takes values between 0 and 1. None of them are 0 or 1.
>
> > Q6: Why does a symmetric situation arise?
>
> A6: This is because we plot the **absolute value** of the noise $|\epsilon_i|$, which is the magnitude of the noise for clear visualization.
>
> > Q7: I did not understand the discussion after Thm 1 (lines 145-183).... Please provide a detailed explanation, as I do not understand the purpose, conclusion, or the meaning conveyed by Figure 2 in this section.
>
> A7: This part explains the “adaptive” nature of the noise. Eq. 12 gives the exact expression for the noise and Fig. 2 visualizes it as the function of the posterior mean. We can rewrite this to make it clearer.
>
> > Q8: What is the role of Section 3.3? How does it help in deriving the conclusion of Section 3.4?
>
> A8: Section 3.3 generalizes Section 3.2 to show that Newton's method provides more adaptive label smoothing. There is no direct connection with Section 3.4.
>
> > Q9: The proof appears to heavily rely on the linear term −yf(x)  in the loss function. Does the result hold for other loss functions as well? Can the class of loss functions discussed by the authors encompass common loss functions such as cross-entropy loss? What effect do they have on different losses (such as different formats in exponential-family loss and some other loss functions beyond exponential-family loss)?
>
> A9: Yes, the results hold for other loss functions including the cross entropy loss as explained in line 205. Please note that the linear term -y f(x) is not a limitation. All exponential-family losses have this term and most of the loss functions used in deep learning are derived from exponential-family losses.

---

> > ### Comment · Reviewer_KjjK · 2025-04-04
> >
> > Thanks for the rebuttal. Unfortunately, the rebuttal has reinforced, rather than resolved, my concerns.
> > - The author's response to the last question did not satisfy me. If the author's conclusion only holds for loss function with linear terms, then this conclusion is too trivial, because taking the gradient of a linear term itself will yield a y. The explanation in line 205 does not demonstrate that this loss function encompasses common loss functions. Additionally, the author did not provide an explanation of how common loss functions are included within the exponential family losses.
> >
> > - In Section 3.4, the author should provide a theoretical derivation for the multi-sample scenario, as I believe approximating all \nabla f_i(\theta^k) as \nabla f_i(\theta_t) is problematic. The error estimation I expect is the discrepancy between this approximation and the true expectation.

---

### Official Review · Reviewer_gwdL · 2025-03-07

**Overall Recommendation:** 2

**Summary:**

The authors demonstrate that variational learning induces adaptive label smoothing and derive the exact form of label noise for various problems. They show that variational learning associates the high noise to atypical samples. Experimental results show that the proposed approach outperforms label smoothing.

**Claims And Evidence:**

The claims made in the submission are supported by clear and convincing evidence.

**Essential References Not Discussed:**

The authors discuss label smoothing but do not include the important paper [a].

[a] Wei, J., Liu, H., Liu, T., Niu, G., Sugiyama, M., & Liu, Y. (2022, June). To Smooth or Not? When Label Smoothing Meets Noisy Labels. In International Conference on Machine Learning (pp. 23589-23614). PMLR.

**Experimental Designs Or Analyses:**

The results showed are interesting and validate the thesis of the paper. However, the authors show few experimental comparisons with alternative literature. For instance, they should include the label smoothing technique proposed in [a].

In addition, they should include comparisons with other adaptive label smoothing techniques that are already cited in the paper, like Ko et al. (2023) and Park et al. (2023).

[a] Wei, J., Liu, H., Liu, T., Niu, G., Sugiyama, M., & Liu, Y. (2022, June). To Smooth or Not? When Label Smoothing Meets Noisy Labels. In International Conference on Machine Learning (pp. 23589-23614). PMLR.

**Methods And Evaluation Criteria:**

The proposed methods make sense for the problem at hand. However, the authors show little comparison with other existing literature.

**Other Comments Or Suggestions:**

- The lines 84-89 in pp. 2 are very similar to lines 35-40 in pp. 1.
- Remove "is" from "The gradient of KL is also simplifies to" in line 161
- The "regularizer" instead of "regularize" in line 193
- "The result shows that the GD steps to optimze Eq. 6 is equivalent to those ...", where "steps" is plural, "is" is singular, and "those" is plural again

**Other Strengths And Weaknesses:**

Strengths
- Interesting results and contributions, as the authors show that variational learning implicitly induces adaptive label smoothing, thus providing an elegant approach for adaptive label smoothing.
- Incremental result on an already published paper, thus enlarging the possible applications of an already existing algorithm, which has already shown its validity in a different context.

Weaknesses
- The authors do not provide the code.
- Too few comparisons with other techniques for classification in the presence of label noise.
- The paper is not well written and contains several grammatical mistakes and typos.

**Questions For Authors:**

- What does it mean "similarly to deep learning" in the first line of pp. 3?
- Is there a motivation why $q(\theta)$ is set to a Gaussian?
- Is there some reference defining what variational learning is? You should cite it before equation (6).
- Can you report all the steps for the proof of Theorem 1?
- Why in the proof of Theorem 1 do you write that the gradient of the KL simplifies to an equation which is instead reported to be the KL itself? Do you mean that the gradient of the KL is the gradient of the squared norm of $\theta_t$? How do you use the gradient of the KL in the proof?
- Can you provide the code?
- Can you provide more experiments as described in the "Experimental Designs Or Analyses" section?

**Relation To Broader Scientific Literature:**

The contributions are incremental to the existing literature as this paper deeply relies on IVON (Shen et al., 2024). However, in this specific case, I do not see this as a negative aspect. On the contrary, I consider it as a strength: IVON was proposed as an optimization method in alternative to Adam. On the contrary, this paper provides a contribution to the label noise literature. The paper by Shen et al. and this paper have in common the usage of variational learning and IVON. But, in my opinion, the contribution of this paper to the noisy labels literature constitutes novel results by relating variational learning and IVON with the label noise setting, thus demonstrating the validity of such an approach.

**Theoretical Claims:**

I checked the proofs. Please see the questions section.

---

> ### Author Rebuttal · Authors · 2025-04-01
>
> We thank the reviewer for their review. As per reviewer's suggestion, we have made the code available at the following [anonymous link](https://drive.google.com/file/d/18l5kuKmmN0H9dud4QPBI3LBeF6XobYC5/view?usp=sharing). We also thank the reviewer for suggestions regarding writing. We will fix these in the next version. Below, we respond to other comments.
>
> > Q1: The authors… do not include the important paper [by Wei et al. 2022].
>
> A1: Thanks for this. We will make sure to cite it in the next version.
>
> > Q2: [The authors] should include comparisons with other adaptive label smoothing techniques that are cited in the paper, like Ko et al. (2023) and Park et al. (2023).
>
> A2: We did not compare to them because these papers are similar to Zhang et al. (this is stated in line 56-62) which we compared to in Fig. 4. We will add these other works but do not expect the results to change due to the similarity of these methods.
>
> > Q3: Is there a motivation why $q(\theta)$ is set to a Gaussian?
>
> A3: There is no specific reason but the Gaussian case is simpler to explain. As shown in Fig. 1, the weight uncertainty in variational learning is expected to improve overconfidence in the predictions. The result is valid for all sorts of variational distributional forms.
>
> > Q4: Is there some reference defining what variational learning is?
>
> A4: We will add a citation of this work: D. M. Blei, A. Kucukelbir, and J. D. McAuliffe. Variational inference: A review for statisticians. Journal of the American statistical Association, 112(518):859–877, 2017.
>
> > Q5: Can you report all the steps for the proof of Theorem 1?
>
> A5: Yes, we will fix this in the camera ready version.
>
> > Q6: Why in the proof of Theorem 1 do you write that the gradient of the KL simplifies to an equation which is instead reported to be the KL itself? Do you mean that the gradient of the KL is the gradient of the squared norm of ? How do you use the gradient of the KL in the proof?
>
> A6: Perhaps the sentence is confusing. What we mean is that the expression for KL is simply the squared norm, whose gradient takes a simple form.

---

> > ### Comment · Reviewer_gwdL · 2025-04-02
> >
> > Thank you for the answers and for providing the code. Unfortunately, I see that you did not report additional experiments as suggested. You claim that Ko et al. (2023) and Park et al. (2023) would achieve similar results to Zhang et al., but I need to see the experiments. In addition, I also want to see a numerical comparison with Wei et al. 2022.
> > Finally, you did not provide the complete proofs of Theorem 1 and Theorem 2 (asked by another Reviewer). For these reasons, I will decrease my final score from 3 to 2. I am still prone to increase it again, if I will receive an appropriate answer reporting all the requested comparisons and proofs.

---

> > > ### Author Response · Authors · 2025-04-02
> > >
> > > We appreciate the acknowledgement of the review. As you may know that this year we have a 5000 character limit for response, which is why our responses are brief; of course it is not a problem for us to write a complete proof. But given the character limit we promised to update it. We could still write it if you think it will help you to reconsider. The proofs are not wrong, just that some steps are skipped to save space.
> > >
> > > Regarding the lack of experiments, the rebuttal period for ICML was just 6 days which is not enough to finish any meaningful experiments, so we decided to tell you the reason why we didn't do them.
> > >
> > > In any case, we hope you understand that the omission of these points are not due to a lack of consideration for your feedback but the short duration and space limit for the rebuttal.
> > >
> > > We thank you nevertheless for your time.

---

### Official Review · Reviewer_XuYB · 2025-03-10

**Overall Recommendation:** 1

**Summary:**

The paper studies the connection between variational inference and adaptive label smoothing. Throughout a thorough derivation and analysis on linear models, the authors show that learning with a variational posterior is equivalent to conventional point estimation with adaptive smooth labels. The paper also extends the analysis to non-linear models (i.e., neural networks) via Taylor's series. Empirical evaluation demonstrates that the smoothness of labels induced through variational inference leads to superior performance compared to other smoothing state-of-the-art methods.

### Update After Rebuttal
The rebuttal does not fully address the concerns initially raised. Hence, I keep the same rating for the paper.

**Claims And Evidence:**

The claim of variational posterior is a type of adaptive label smoothing is clear and convincing for linear models. The extension to deep neural network part is unclear and not convincing.

Another slightly over-claim is the assumption on Gaussian variational posterior. In the current form, it seems that the analysis is applicable when the variational posterior is Gaussian. This is to simplify the calculation of KL divergence. Thus, it may not hold for other kinds of variational posteriors.

**Essential References Not Discussed:**

I am not aware of any prior study in a similar line of research.

**Experimental Designs Or Analyses:**

To my understanding, the empiricial evaluation is to compare IVON with other label smoothing methods, where prediction accuracy of models trained on those smooth labels is used as a proxy of performance. If this is the case, there is a confounding factor here. In particular, variational inference is often more robust than point-estimate. IVON is indeed a method to learn a variational distribution of model parameters by minimizing the variational-free energy (i.e., Eq. (6)), while other methods are point estimates. Hence, using such evaluation benchmark to demonstrate may be insufficient because it is not clear the superior performance is due to adaptive label smoothing or the modeling using variational inference.

**Methods And Evaluation Criteria:**

The empirical evaluation presented in the paper is reasonable only if the claim is correct. If it is not correct, then the IVON is a variational inference learning which learns the distribution of model parameters, while the other baselines are point-estimates trained with smooth labels. In that case, it is not apple-to-apple comparison.

**Other Comments Or Suggestions:**

Typos:
 - Line 171: atypical => a typical

**Other Strengths And Weaknesses:**

N/A

**Questions For Authors:**

Two main questions should be addressed to improve the contributions and effectiveness of the analysis studied in the paper:
 - Make it clearer for the approximation in subsection 3.4: in its current form, the approximation is quite rough and limited.
 - Another benchmark to demonstrate while avoiding the confounding factor due to Bayesian inference on model parameters.

**Relation To Broader Scientific Literature:**

The idea of the paper is interesting because it bridges the knowledge between variational inference (or could be Bayesian inference in general) to label smoothing. If that is correct, the insight from this study is valuable in terms of understanding.

**Theoretical Claims:**

The theoretical claim in the paper is correct for linear models in general. However, in the derivation for neural networks, that may not hold. The reason is that the analysis in sub-section 3.4 is too rough in terms of approximation. In particular, the approximation of the gradient at line 253 is a zero-order one.

---

> ### Author Rebuttal · Authors · 2025-04-01
>
> We thank the reviewer for their review. The reviewer found that “the extension to deep neural network part is unclear and not convincing” and also that the assumption on Gaussian variational posterior to be “slightly over-claim”. We do not agree with the reviewer's assessment and challenge it below.
>
> First of all, **the suggestion that the result “may not hold for other kinds of variational posteriors” is incorrect**. We consider the Gaussian case because it is easy to show these connections, not because the calculation of KL divergence simplifies. In fact, KL divergence for any exponential family has a closed-form expression; e.g., see Page 15 onwards in [Niesen and Gacia]( https://arxiv.org/pdf/0911.4863) for such a list. In general, weight-uncertainty (or perturbation) in variational learning is expected to have a positive effect on the overconfidence of the label. Our paper shows cases where the connection is clear and it is not an overclaim.
>
> Second, the reviewer also suggests that our empirical evaluations are not “apple-to-apple” comparisons because we compare variational learning to point estimates. This does not make much sense to us because such comparisons are routinely used in Bayesian deep learning, where the performance of the mean of the Gaussian posterior is compared to the point estimates. We request the reviewer to reconsider their assessment.
>
> Third, the reviewer is critical of the analysis in Sec 3.4 calling it “too rough in terms of approximation”. Specifically, they dismiss the “approximation of the gradient at line 253” because it is a zeroth order approximation. This is not true because the result is simply a consequence of the first-order approximation, as shown below.
> $f_i(\theta) \approx f_i(\theta_t) + \nabla f_i(\theta_t) (\theta - \theta_t)
> \implies
> \nabla f_i(\theta) \approx \nabla f_i(\theta_t)$
>
> Finally, the reviewer dismisses the experiments stating that “variational learning is often more robust than point-estimates” but they ignore the fact that we are comparing a point estimate **with label smoothing** to variational learning **without** an explicit label smoothing. For some reason, the reviewer then claims this to be “confounding” stating that “it is not clear if the superior performance is due to adaptivity label smoothing or variational inference”. This makes little sense to us because our main result is to show that variational learning induces the adaptive label smoothing. These two things are the same thing and there is no issue of one confounding the other.
>
> We would also like to say that “atypical” is not a typo (as pointed out by the reviewer). We really mean to say that the higher noise is assigned to examples are **not** typical.

---

> > ### Comment · Reviewer_XuYB · 2025-04-06
> >
> > Thank you the authors for replying to the concerns I raised in the initial review.
> >
> > For the comparison, the rebuttal is still unclear and has not addressed my concern. In my intial review, I mentioned that the proposed method is a variational inference method, meaning that it is to learn the distribution of model parameters, while the methods used in the comparison are point-estimate (i.e., Dirac Delta posterior). Hence, it is expected to have something to compare. In addition, as mentioned in the initial review, if the claim about "adaptive smoothing label" is incorrect, applying variational inference can also be compared with other methods and show (properly) better performance. Unfortunately, the authors did not clarify for this point.
> >
> > For the approximation, I apologize for not making it clear. It is properly not zero-order, but the approximation assumes that the function $f_{i}(\theta)$ is linear w.r.t. $\theta$. That is why the gradient is approximated to be a constant (e.g., $\Delta f_{i} (\theta_{t})$). Such an approximation is not well justified and hence, my concern.
> >
> > And thank for clarifying to me about "atypical". I am not aware of that word until now, but glad to learn a new way to express something not common.

---

### Official Review · Reviewer_A2DP · 2025-03-16

**Overall Recommendation:** 2

**Summary:**

This paper establishes a relationship between variational learning and adaptive label smoothing. It introduces the validation of this relationship across three models: Logistic Regression, Generalized Linear Model, and neural networks optimized using the IVON optimizer. The results show that directly learning the posterior using a variational method naturally yields an adaptive label smoothing. The proposed method is compared with other label smoothing techniques and the SAM method. The findings indicate that on mainstream datasets such as CIFAR-10/100 and Clothing1M, the proposed method achieves or surpasses the performance of these methods.

**Claims And Evidence:**

Yes.

**Essential References Not Discussed:**

No.

**Experimental Designs Or Analyses:**

1. The Bayesian version of label smoothing is evaluated on the task of learning with noisy labels. However, this is not comprehensive. Other tasks such as out-of-distribution (OOD) detection and calibration should also be included for a more thorough evaluation.

2. More recent baselines need to be incorporated for comparison, as the latest baseline referenced in the study was published in IEEE TIP 2021.

**Methods And Evaluation Criteria:**

MNIST, CIFAR-10/100, and Clothing1M are widely utilized benchmarks in the field of label noise robust learning. To ensure comprehensive validation, it would be beneficial to incorporate additional classification benchmarks, such as Stanford Flowers, into the evaluation framework. This inclusion would provide a more thorough assessment of the proposed method's efficacy across diverse datasets.

**Other Comments Or Suggestions:**

No.

**Other Strengths And Weaknesses:**

Please refer to the Broader Scientific Literature.

**Questions For Authors:**

No.

**Relation To Broader Scientific Literature:**

The study establishes a connection between variational learning and the adaptive label smoothing strategy, particularly in the context of instance-specific label noise. While this is an interesting contribution, I believe it may not be sufficient for an ICML-level publication. The work could benefit from deeper theoretical insights, broader empirical validation across diverse tasks (e.g., OOD detection, calibration), and comparisons with more recent and state-of-the-art baselines to strengthen its impact and novelty.

**Theoretical Claims:**

I have thoroughly examined the theoretical claim, and it has been confirmed to be correct.

---

> ### Author Rebuttal · Authors · 2025-04-01
>
> We thank the reviewer for their review. The reviewer finds that the work “may not be sufficient for an ICML-level publication” due to lack of “deeper theoretical insights” and “broader empirical validation”. **We do not agree with the assessment. Our connections between variational learning and label smoothing are entirely new and precisely derived for generalized linear models.** Through approximations, we show that they are expected to hold for neural networks as well (Sec 3.4) which is further validated empirically.
>
> The reviewer has asked to include “other tasks such as out-of-distribution (OOD) detection and calibration” and “more recent baselines”, but we do not think that these add much value. This is because we are not claiming that variational learning induces a **better** smoothing. Instead, the goal of the experiments is to validate the approximations used in Sec 3.4 and confirm the induced nature of adaptive label-smoothing of variational learning. This is why we show similarities to Zhang et al.’s method in Fig. 4.
>
> The reviewer also suggested “to incorporate additional classification benchmarks, such as Stanford Flowers” but they did not provide an exact source for this dataset and we were unable to find it by ourselves. We will be happy to include it if the reviewer can provide a citation or link.

---

### Decision · Program_Chairs · 2025-05-01

**Decision:**

Reject

**Comment:**

The paper discusses a view of variational learning as “adaptive” label smoothing for generalized linear models (and in turn for neural networks with some approximations). To that effect, the paper claims, variational learning is an effective mechanism to adaptively apply label smoothing for robustness to labelling error and data shift. The label smoothing effect is empirically verified and the robustness is shown to be empirically superior to standard label smoothing and on a par with an existing adaptive method.

The paper was reviewed by a panel of four experts in the different aspects of the paper including variational learning, label smoothing and robustness to label noise. They were initially divided in their judgement, with three leaning to the reject side (ratings 1, 2, 2) and one leaning to accept (rating 3).

The reviewers appreciate the relevance of the drawn connection between variational learning and label smoothing and the characterization of the adaptivity form.

On the other hand, the reviewers raised concerns regarding (1) the extent of the experiments (more datasets, more tasks e.g., OoD robustness, and more baselines), (2) the required approximations to elevate the formal result to the case of neural networks, (3) the conclusivity of experiments in isolating the label smoothing effect of variational learning from any other potentially-helpful property that it brings, and (4) several unclarities in the presentation.
The rebuttal was attended by all reviewers and discussed with some reviewers. The final ratings, after the rebuttal, have all reviewers leaning towards reject with the ratings of 1, 2, 2, 2.

The AC believes the formalised connection is intuitive and worth disseminating in a top ML conference such as ICML. The AC also does not see a need for an increase in the number of datasets or tasks. The AC further finds the approximations used for the case of neural networks acceptable and common. However, a major revision seems to be required to address the questions, unclarities, and at times misconceptions caused for all reviewers reading the current version. Furthermore, the empirical results can be extended to (1) isolate the label smoothing effect of variational learning as *the* reason behind the improvement as claimed by the earlier parts of the paper and the formal results, this for instance can entail including an experiment with no erroneous labels (and perhaps no “atypical” sample) to see whether significant improvement is not anymore observed, and (2) include more existing adaptive label smoothing techniques as baselines, which is important since the paper’s main contribution is taking variational learning as adaptive label smoothing. The AC does agree with the authors that the paper would not need to improve those methods but they are needed for a proper and full context and potential follow-up discussions as to the reasons behind any potential differences in performances.

Therefore, while the AC acknowledges the merits, it does not overrule the unanimous suggestion of the four reviewers due to the aforementioned shortcomings and recommends rejection.